# Lamivudine, Doravirine, and Cabotegravir Downregulate the Expression of Human Endogenous Retroviruses (HERVs), Inhibit Cell Growth, and Reduce Invasive Capability in Melanoma Cell Lines

**DOI:** 10.3390/ijms25031615

**Published:** 2024-01-28

**Authors:** Valentina Zanrè, Francesco Bellinato, Alessia Cardile, Carlotta Passarini, Jacopo Monticelli, Stefano Di Bella, Marta Menegazzi

**Affiliations:** 1Section of Biochemistry, Department of Neuroscience, Biomedicine and Movement Sciences, University of Verona, Strada Le Grazie, 8, 37134 Verona, Italy; valentina.zanre@univr.it (V.Z.); alessia.cardile@univr.it (A.C.); carlotta.passarini@univr.it (C.P.); 2Section of Dermatology and Venereology, Department of Medicine, University of Verona, Piazzale Stefani 1, 37126 Verona, Italy; francesco.bellinato@univr.it; 3Infectious Diseases Unit, Trieste University Hospital (ASUGI), Piazza dell’Ospitale 1, 34129 Trieste, Italy; jacopo.monticelli@asugi.sanita.fvg.it; 4Clinical Department of Medical, Surgical and Health Sciences, University of Trieste, Piazzale Europa 1, 34127 Trieste, Italy; sdibella@units.it

**Keywords:** HERV-K, antiretroviral drugs, invasion, apoptosis, ferroptosis, PD-L1 expression, STING, dacarbazine

## Abstract

This study explores the impact of antiretroviral administration on the expression of human endogenous retroviruses (HERVs), cell growth, and invasive capability of human melanoma cell lines in culture. We investigated three antiretrovirals—lamivudine, doravirine, and cabotegravir—in A375, FO-1, and SK-Mel-28, BRAF-mutated, and in MeWo, P53-mutated, melanoma cell lines. The findings indicate a general capability of these drugs to downregulate the expression of HERV-K Pol and Env genes and hinder cell viability, mobility, and colony formation capacity of melanoma cells. The antiretroviral drugs also demonstrate selectivity against malignant cells, sparing normal human epithelial melanocytes. The study reveals that the integrase inhibitor cabotegravir is particularly effective in inhibiting cell growth and invasion across different cell lines in comparison with lamivudine and doravirine, which are inhibitors of the viral reverse transcriptase enzyme. The investigation further delves into the molecular mechanisms underlying the observed effects, highlighting the potential induction of ferroptosis, apoptosis, and alterations in cell cycle regulatory proteins. Our findings showed cytostatic effects principally revealed in A375, and SK-Mel-28 cell lines through a downregulation of retinoblastoma protein phosphorylation and/or cyclin D1 expression. Signs of ferroptosis were detected in both A375 cells and FO-1 cells by a decrease in glutathione peroxidase 4 and ferritin expression, as well as by an increase in transferrin protein levels. Apoptosis was also detected in FO-1 and SK-Mel-28, but only with cabotegravir treatment. Moreover, we explored the expression and activity of the stimulator of interferon genes (STING) protein and its correlation with programmed death-ligand 1 (PD-L1) expression. Both the STING activity and PD-L1 expression were decreased, suggesting that the antiretroviral treatments may counteract the detrimental effects of PD-L1 expression activation through the STING/interferon pathway triggered by HERV-K. Finally, this study underscores the potential therapeutic significance of cabotegravir in melanoma treatment. The findings also raise the prospect of using antiretroviral drugs to downregulate PD-L1 expression, potentially enhancing the therapeutic responses of immune checkpoint inhibitors.

## 1. Introduction

Cutaneous melanoma incidence has consistently risen over the past few decades in various susceptible populations [1]. Melanoma is also the most invasive form of skin cancer, and patients with advanced metastatic melanoma face a high risk of death, with a median survival rate of less than one year [2,3]. Melanoma initiation is driven in melanocytes by various genetic and epigenetic events prompted by adverse environmental conditions, particularly ultraviolet (UV) radiation. These events are further intensified by a sustained pro-inflammatory response of the tissue [4]. Interestingly, the same insults—chemicals, UV radiation, and endogenous injurious factors such as cytokines and inflammatory mediators—can also induce the expression of mobile genetic elements derived from ancestral retroviruses [5,6,7]. Therefore, it is not surprising that these elements are now recognized to support both melanoma initiation and progression [8,9,10].

Human endogenous retroviruses (HERVs) are single-stranded RNA viruses that integrate into the human germline through their long terminal repeats (LTRs) [11]. By replicating through RNA intermediates and disseminating within genomes, they constitute up to 8% of the human genome [12]. Nevertheless, most of them exist as remnants of proviruses represented by solitarily LTRs. About 1.5% of HERV loci carry a complete genome consisting of two LTRs flanking a set of internal retroviral genes, such as capsid (Gag), protease (Pro), polymerase (Pol), and envelope (Env) [11,13,14]. Additionally, some complete HERVs contain numerous mutations accumulated over evolution, rendering them non-functional and incapable of expressing retroviral proteins. However, the remaining HERV elements are still active in their host genome [15]. HERVs have been classified based on the tRNA type that binds to the viral primer binding site to initiate reverse transcription [16]. Thus, a provirus that uses a lysine (K) tRNA is named HERV-K. This is one of the most studied families of HERVs, as described below.

While under physiological conditions only a few HERV-K loci appear to be active, in melanoma patients as well as in melanoma cell lines, the number of transcribed HERV-K loci becomes significant [17]. A study reported that human melanoma cells producing retrovirus-like particles of the HERV-K family exhibit an active reverse transcriptase (RT) enzyme. The authors also observed the expression of several pro-viral proteins in cells from human primary or metastatic melanoma, unlike normal melanocytes or lymph nodes [18]. Buscher et al. further demonstrated that 45% of melanoma biopsies and 44% of melanoma cell lines expressed Env or other HERV-K proteins, and antibodies against HERV-K were frequently detected in the serum of patients with melanoma [19].

Importantly, HERV-K expression can directly influence melanoma cell proliferation, differentiation, and survival [20,21].

Identifying appropriate therapeutic strategies for the treatment of melanoma has consistently posed challenges, given its resistance to standardized approaches, frequent relapses, and low immunogenicity [2]. Nevertheless, in the last decade, the utilization of targeted therapies and immunotherapies has significantly improved the prognosis of melanoma [2].

Approximately 40–60% of melanomas harbor oncogenic mutations in the BRAF kinase gene. In patients with BRAF-mutated melanoma, the downstream pro-survival mitogen-activated protein kinase (MAPK) pathway is consistently activated, leading to elevated tumor growth and progression [22]. Thus, the primary objective of targeted therapy is to inhibit the function of BRAF kinase by using BRAF-inhibitors or the MAPK pathway with MAPK-inhibitors [22].

In contrast to targeted therapy, systemic immunotherapy is not tailored to specific mutations. Remarkably, reinstating immunological control of tumor growth should be a primary goal of cancer therapy. It is well established that tumors can only originate and progress in the context of failing immune responses [23]. Immune checkpoint inhibitors, including the antibodies that block CTLA-4 (cytotoxic T-lymphocyte-associated protein-4) and PD-1 (programmed cell death protein-1) expressed on lymphocyte membranes, aim to enhance T-cell infiltration in melanoma and stimulate an immune response against the tumor [24]. On the other hand, the tumor cell surface expresses programmed death-ligand 1 (PD-L1). When PD-L1 binds to the PD-1 receptor on T cells, it hinders the immune response, leading to T-cell death [24]. The high PD-L1 expression in cancer cells is indeed associated with a poor prognosis [25]. Unfortunately, pro-inflammatory cytokines such as interferon-γ and interleukin-6 can induce immunotherapy resistance by increasing PD-L1 expression [26,27].

Despite significant progress, resistance to both immunotherapy and targeted therapy inevitably occurs in many cases [28]. These drugs are also burdened by long-term immune-related adverse events (i.e., hypothyroidism, hypophyditis) and cutaneous, cardiac, and gastrointestinal toxicity [29].

The aim of this study is to explore and compare the effects of three antiretroviral drugs that could potentially hinder HERV-K expression and melanoma cell aggressiveness.

Commonly, antiretroviral drugs are classified into a few classes based on their mechanism of action and structure. The most commonly used are nucleoside/nucleotide analogues that act as reverse transcriptase inhibitors (NRTIs), non-nucleoside/nucleotide reverse transcriptase inhibitors (NNRTIs), protease inhibitors (PIs), and integrase strand transfer inhibitors (INSTIs). In particular, INSTIs target the ability of linear double-stranded DNA (dsDNA), derived from retrotranscribed viral RNA, to insert into the host genome using the viral integrase enzyme [30].

To the best of our knowledge, antiviral drugs have not been approved as anti-proliferative agents. However, promising in vitro studies have demonstrated that reverse transcriptase inhibitors can modulate cell growth and differentiation across various cancer types, as reviewed by [31]. For instance, amantadine, ribavirin, and pleconaril, tested against HCT8 colonic cancer cells, were shown to have cytotoxic activity and to force the downregulation of HERV proteins [32]. Moreover, ritonavir, atazanavir, and lopinavir have been shown to reduce the proliferation of schwannoma and grade I meningioma cells [33]. Additionally, nevirapine has demonstrated the ability to reduce cell growth and promote differentiation in acute myeloid leukemia (AML) cells and primary blasts from two AML patients [12]. Landriscina et al. reported reversible inhibition of cell proliferation in an undifferentiated thyroid carcinoma cell line with high endogenous RT activity upon treatment with the NNRTIs nevirapine and efavirenz [34]. These NNRTIs were also tested on CD133-positive melanoma stem cells, showing a downregulation of HERV-K activity concurrent with a decrease in the proliferative rate and CD133 expression [9]. Recently, Giovinazzo et al. compared the effects of the NRTI azidothymidine and the NNRTI efavirenz on cell lines derived from lung adenocarcinoma, hepatocellular carcinoma, and melanoma [35]. The treatments affected HERVs transcriptional activity in parallel with the reduction of cell growth, clonogenic activity, and induction of apoptosis [35].

Given the similarities between human immunodeficiency virus (HIV) and HERVs, antiretroviral drugs are intriguing; millions of people living with HIV/AIDS, generally experiencing negligible side effects, take these drugs daily. In order to be virologically and clinically effective, an antiretroviral therapy against HIV consists of a variable combination of 2–4 drugs belonging to at least two of the before-mentioned classes (for instance, a combination of tenofovir plus emtricitabine—both NRTIs—plus bictegravir, an INSTI) [36]. Despite their efficacy, protease inhibitors (PIs) have recently been discontinued as first-line antiretroviral therapy due to ‘poor practicality’, stemming from frequent interactions with concurrent medications and other issues. To choose a representative from each of the most frequently employed antiretroviral classes, we opted for lamivudine as an NRTI, doravirine as an NNRTI, and cabotegravir as an INSTI.

Lamivudine has been previously investigated as an effective anti-HERV-K agent both in vitro and in a clinical neurological study [37,38]. Lamivudine is also active against the hepatitis B virus (HBV), and it is widely used as a prophylaxis in order to prevent the reactivation of HBV in some cancer patients in conjunction with chemotherapy [39]. With a half-life of 10.5–15.5 h in HIV-infected cell lines, allowing for once-daily dosing, and its relative safety, it has been used as an anti-HIV and anti-HBV agent in millions of people since 1995 [40]. Considering its effectiveness as an anti-HERV-K agent and its clinical safety, we chose lamivudine as a ‘positive control’ agent.

Doravirine, a recently approved NNRTI, is characterized by a high genetic barrier to resistance mutations and greater safety compared to other NNRTIs, such as efavirenz or rilpivirine [41]. Older NNRTIs (such as efavirenz, etravirine, and nevirapine) have been studied as anti-HERV agents with conflicting results, possibly due to the HERV-K reverse transcriptase structure. This structure is more similar to the HIV-2 structure, where these classes of molecules are clinically ineffective. Alternatively, conflicting results may be attributed to constitutive resistance mutations to NNRTIs in HERV-K [42,43]. However, to our knowledge, doravirine has not been studied as an anti-HERV agent, and its ability to inhibit the RT of some NNRTI-resistant viral strains led us to choose this drug as ‘representative’ of NNRTIs [44].

Cabotegravir, a second-generation anti-retrovirus drug that was licensed in early 2021, is a highly potent INSTI with inhibitory concentration 50% (IC50) values of 0.2 nM, effectively inhibiting HIV-1 replication in peripheral blood mononuclear cells in vitro [30]. Intramuscular administration of cabotegravir in patients (in combination with rilpivirine) has been increasingly chosen as an effective antiretroviral agent. This is attributed to its half-life ranging from 5.6 to 11.5 weeks, allowing for long-acting regimens that can be injected monthly or bimonthly [45]. Due to its relative clinical safety and its extremely long half-life, cabotegravir has also been studied as a pre-exposure prophylactic agent in order to prevent HIV infection, with promising results [46]. Therefore, what led us to select cabotegravir as a ‘representative’ of INSTIs was its interesting half-life. To our knowledge, cabotegravir has not been studied as an anti-HERV agent. In contrast, other INSTIs, such as raltegravir and elvitegravir, have been previously tested as anti-HERV-K agents, yielding promising results [37].

All these antiviral drugs were tested on three BRAF-mutated and one P53-mutated melanoma cell lines to assess their ability to hinder HERV-K gene expression, melanoma cell growth, mobility, and colony formation. Additionally, their capacity to modulate the expression of proteins influencing tumor growth and progression was also investigated.

## 2. Results

### 2.1. Influence of Antiretrovirals on the Sulforhodamine B (SRB) Cell Viability Assay

In our investigation, we selected the A375, SK-Mel-28, and FO-1 melanoma cell lines due to their possession of the BRAF^V600E^ mutation, a genetic alteration observed in 50–60% of melanoma patients. Cells carrying the BRAF^V600E^ mutation manifest heightened aggressiveness and accelerated growth rates when contrasted with cells lacking this mutation. Additionally, these mutated cells showcase the preferential activation of distinct oncogenic pathways [47]. Furthermore, to validate our observations in melanoma cells lacking the BRAF mutation, in some experiments we examined the behavior of the MeWo cell line. This cell line possesses a BRAF-wild type but a p53-mutated genotype, serving as a pertinent model for investigating the impact of p53 mutations in the absence of BRAF mutations.

At first, our aim was to investigate whether the antiretroviral drugs exerted cytotoxic or cytostatic effects on melanoma cells. To assess the viability of SK-Mel-28, A375, FO-1, and MeWo cell lines, we conducted a Sulforhodamine B assay (SRB). Preliminary experiments were carried out at different time points (24, 48, and 72 h) following antiretroviral drug administration. Various concentrations, derived from literature studies, were initially tested for each of the three treatments. The impact of these drugs, already used in vivo, was assessed on normal human epithelial melanocytes (NHEM) to evaluate their non-toxic effects on the viability of normal skin cells. Based on these preliminary experiments, the optimal non-toxic ranges were determined for lamivudine (0.05 to 5 µM), doravirine (0.5 to 10 µM), and cabotegravir (0.1 to 5 µM). For other investigations, two concentrations were chosen for each antiretroviral drug, specifically lamivudine at 0.1 and 1 µM, doravirine at 1 and 5 µM, and cabotegravir at 0.5 and 3 µM. Dacarbazine (DTIC) was chosen for the comparison of antiretrovirals with a medication commonly employed in the treatment of melanoma. DTIC is an FDA-approved antineoplastic alkylating agent effective against metastatic melanoma, regardless of the presence of the BRAF mutation [48]. DTIC was used in the culture medium in the range of 0.5–20 µM.

As illustrated in Figure 1, none of the antiretroviral treatments exhibited any discernible effect on NHEM. Conversely, among all the treatments, cabotegravir appeared to possess the greatest capability to diminish cell viability across all four melanoma cell lines. Lamivudine and doravirine seem to exert a milder effect on melanoma cell viability (Figure 1). Surprisingly, DTIC exhibits a comparable effect on melanoma cell viability as lamivudine or doravirine, albeit to a lesser extent than cabotegravir (Figure 1).

### 2.2. Effects of Lamivudine, Doravirine, and Cabotegravir on HERV-K Gene Expression

In order to investigate the capability of these three antiretroviral drugs in mitigating the expression of human endogenous retroviral-K genes (HERV-K genes), real-time PCRs were performed to determine the mRNA levels of Pol and Env genes in SK-Mel-28, A375, FO-1, and MeWo cell lines.

Initially, bibliographic research was conducted to identify potential housekeeping genes for normalization purposes. The evaluated genes included glyceraldehyde-3-phosphate dehydrogenase, TATA-Box binding protein, cyclophilin A, RNA polymerase II subunit A, 60s ribosomal protein L32, beta-2-microglobulin, β-actin, receptor for activated C kinase 1, 18S ribosomal RNA (18S rRNA), H2A Clustered Histone 11, E-cadherin, tyrosine-protein kinase receptor UFO, and cluster of differentiation 151 (CD151). Among these, only CD151 and 18S rRNA were found to remain unaltered in response to the antiretroviral drugs. Finally, CD151 was selected as the preferred housekeeping gene for normalizing the expression of HERV-K genes, as its cycle threshold values (Ct) closely resembled those of the viral genes.

Despite the variability observed across the four experiments conducted in each cell line and treatment, our findings indicate a trend toward a decrease in the expression of both Pol and Env genes following 1 µM lamivudine, 5 µM doravirine, and 3 µM cabotegravir in SK-Mel-28, A375, FO-1, and MeWo cell lines (Figure 2).

### 2.3. Effects of Lamivudine, Doravirine, and Cabotegravir on Melanoma Cell Migration Capability

Cell migration ability can be evaluated by a wound healing assay. This methodology consists of performing an artificial scratch in a confluent cell monolayer and quantifying the rate at which cells can close this wound over time.

In Figure 3, we can appreciate the ability of 1 µM lamivudine, 5 µM doravirine, and 3 µM cabotegravir to slow down cell migration. In SK-Mel-28, only lamivudine and cabotegravir were able to reduce the wound closure significantly (t48 ** p <* 0.05) compared to the controls. Instead, in the A375 cell line, doravirine and cabotegravir were found to be the most effective drugs (t24, t32 *** p <* 0.01; t48 ** p <* 0.05). In the case of FO-1, we can appreciate a non-significant bland delay in the wound closure with 1 µM lamivudine and 5 µM doravirine. Moreover, with the presence of cabotegravir in cell culture medium, we were able to perceive a significant (t48 ** p <* 0.05) abate of cell migration compared to the control. In the wound healing assays performed with MeWo cell lines, in treated and non-treated samples, we could notice a much slower cell migration capability compared to the other melanoma cell lines. Nevertheless, the differences between control and treatments were significant only at 48 h with all three drugs (t48 ** p* < 0.05). Notably, the treatment with cabotegravir was the most effective in slowing down cell migration in all cell lines. DTIC was employed for comparison, and its effects on cell migration are reported in Figure 3. DTIC was effective in slowing down cell migration in A375 (t24, t36, and t48 ** p* < 0.05), FO-1 (t24, and t32 ** p* < 0.05), and SK-Mel-28 (t48 *** p* < 0.01) cell lines. On the flip side, MeWo cell mobility remained unaffected by DTIC (Figure 3).

### 2.4. Effects of Lamivudine, Doravirine, and Cabotegravir on the Ability of SK-Mel-28, A375, FO-1, and MeWo Cells to Form Colonies in Soft Agar

The potential of retroviral drugs to hamper melanoma cell growth in an anchorage-independent manner was evaluated by a soft agar colony formation assay. Lamivudine, doravirine, and cabotegravir, at concentrations of 1, 5, or 3 µM, respectively, were administered to the four melanoma cell lines. Figure 4 presents both the quantification of colony number and representative pictures of colony size and density for each antiretroviral treatment and the untreated control. Despite lamivudine and doravirine demonstrating proficient efficacy in reducing the number of colonies in A375 and FO-1, their impact on SK-Mel-28 and MeWo cell lines appears relatively modest, with values not reaching statistical significance (Figure 4). Instead, cabotegravir was found to be the most effective treatment, leading to a massive reduction in both colony size and number in A375 and FO-1 cell lines and preventing colony formation in SK-Mel-28 and MeWo cell lines (Figure 4)

### 2.5. Lamivudine, Doravirine, and Cabotegravir Affect the Expression Levels of STING and PD-L1

We conducted immunoblot analyses to assess the expression levels of proteins involved in immune response regulation. Our analysis focused only on the three cell lines with BRAF mutations to ensure greater homogeneity.

The stimulator of interferon genes (STING) protein undergoes phosphorylation in response to bacterial and viral infections [49]. Immunoblots revealed elevated phosphorylation levels of STING (pSTING) in control samples (Figure 5). Following the administration of antiretrovirals for 24 or 48 h, the expression of pSTING decreased, while total STING protein levels remained unaffected. Simultaneously, the levels of PD-L1 expression also decreased (Figure 5). Importantly, PD-L1 expression can be induced by inflammatory cytokines, such as γ-interferon, which is, in turn, upregulated by activated STING [26,27].

### 2.6. Lamivudine, Doravirine, and Cabotegravir Affect the Expression Levels of Proteins Regulating Cell Cycle and Death

The cell cycle is governed by various cyclins and the activation of cyclin kinases. Specifically, the G1-S checkpoint is overseen by cyclin D1 and the hyperphosphorylation of retinoblastoma protein (pRb). Our results revealed a reduction in the expression of cyclin D1 and in the phosphorylation of the Rb protein after 24 h of retroviral treatments in SK-Mel-28 and A375, while no such effect was observed in the FO-1 cell line (Figure 6). 

The reduction of cell viability, as previously attested by the SRB assay, may be explained by a decrease in cell proliferation and/or an increase in cell death. Therefore, we assessed the activation of apoptotic cell death through the expression of the cleaved form of the poly (ADP-ribose) polymerase (cPARP), which is cleaved by effector caspases during apoptotic activation. Additionally, we examined the presence of ferroptosis through the decreased expression of glutathione peroxidase 4 (GPX-4) and ferritin (FTH-1) and/or an increase in transferrin. FTH-1 and transferrin regulate intracellular free iron content, while GPX-4 protects cell membranes from lipid peroxidation [3].

Immunoblot results indicate a substantial increase in cPARP expression levels in FO-1 and SK-Mel-28 cells, specifically associated with cabotegravir treatment (Figure 7). Conversely, no increase in apoptosis beyond basal levels was observed in A375 cell lines. A significant decrease in GPX-4 and/or FTH-1 expression levels was noted in all three BRAF-mutated melanoma cell lines, with a particularly pronounced effect seen with cabotegravir treatment (Figure 7). An elevated expression of transferrin, responsible for transporting iron into cells, may amplify the occurrence of ferroptosis.

It should be noted that our findings do not preclude the possibility of other types of cell death occurring at the same time.

The expression levels of metalloproteinase 2 are slightly decreased in the BRAF-mutated cell lines (Figure 7). This observation aligns with the previously reported reduction in cell mobility and colony formation.

## 3. Discussion

Previous studies have demonstrated that HERV-K expression within neurons of patients with amyotrophic lateral sclerosis may contribute to neurodegeneration and disease pathogenesis [50]. HERVs have been found to be upregulated in several tumor types, particularly in melanoma [17,18,19]. Notably, Krishnamurthy et al. engineered T-cells using genetic modifications to target HERV-K in highly malignant melanoma cells, resulting in a substantial antitumor effect [51].

Despite the fact that no drugs are specifically designed to target HERVs, a range of pharmaceutical molecules have been developed to combat retroviruses, with a particular focus on HIV-1 infection [37]. It is important to note that individuals with HIV/AIDS require long-term support through highly active antiretroviral therapy (HAART), which may influence the occurrence of associated malignancies. The documented incidence of melanoma among people living with HIV/AIDS varies from 16.4 to 175.7 per 100,000 person-years in selected American and European cohorts during the early HAART era [52]. Until the first decade of the current century, most studies, but one, indicated no discernible cumulative trend in melanoma incidence among individuals living with HIV/AIDS compared to the general population [52]. Conversely, there is a notable increase in the risk of melanoma among other immunosuppressed groups who do not undergo antiviral treatment, such as recipients of solid organ and bone marrow transplants [53].

In this scenario, a valid inquiry emerges regarding whether employing antiretroviral drugs to downregulate HERV-K expression could potentially alleviate the malignant features exhibited by melanoma cells.

In the present work, we verified the ability of two RT inhibitors: the NRTI lamivudine and the second-generation NNRTI doravirine, as well as the INSTI cabotegravir, to downregulate the expression of HERV-K Pol and Env genes. Even though, in not all cases, the RT-PCR data reached statistical significance, our findings suggest a generalized capability of lamivudine, doravirine, and cabotegravir to inhibit the expression of HERV-K Pol or Env genes in melanoma cell lines (Figure 2). The variable responses observed among the cell lines to each antiretroviral drug do not appear to be related to other parameters measured in this study, including cell growth, mobility, and colony formation, or to the presence of the BRAF mutation.

No previous literature studies so far have specifically investigated the efficacy of doravirine and cabotegravir in inhibiting HERVs, possibly due to their recent approval. Furthermore, the effectiveness of other antiretrovirals against HERVs remains a topic of debate. Tyagi et al. reported that PIs were less effective than RT or integrase inhibitors in hindering HERV replication [37]. Another study suggests that several NRTIs, including lamivudine, blocked RT activity and consistently inhibited the replication of HERV-K genomes. However, HIV-1-specific NNRTIs, PIs, and INSTIs did not affect HERV-K, except for the NNRTI etravirine [42]. Recently, Baldwin et al. reported in vitro data, indicating that antiretroviral nucleoside analogs, including lamivudine 3-phosphate, exhibit low HERV-K RT inhibition, whereas classic nonnucleoside analogs do not inhibit HERV-K RT at all [43]. The discrepancies observed among these studies may be attributed to variations in the methodology used to assess retroviral effectiveness, all of which were conducted in vitro. In our investigations, we specifically measured their effectiveness in cells. Notably, our data align with findings from the TVM-A12 melanoma cell line treated with azidothymidine, efavirenz, or nevirapine, albeit under different experimental conditions [9,35].

To assess the impact of lamivudine, doravirine, and cabotegravir against melanoma cell growth, cell viability assays revealed a concentration-dependent reduction in cell growth for all tested drugs (Figure 1). Remarkably, each antiretroviral drug, at the same concentration used in melanoma cells, did not significantly reduce cell viability in NHEM (Figure 1), suggesting their peculiar selectivity against malignant cells. No different responses to each treatment were visualized between A375, FO-1, SK-Mel-28 BRAF-mutated cell lines, and the MeWo cell line, which harbors a BRAF-wild type but a p53-mutated genotype (Figure 1). However, the integrase inhibitor cabotegravir was more effective compared to lamivudine, doravirine, and the antineoplastic alkylating agent DTIC in reducing cell viability (Figure 1). Our findings align with another study that illustrates in vitro treatment with azidothymidine or efavirenz. This study showcases the capability of these treatments to downregulate HERV-K expression and inhibit cell growth in TVM-A12 melanoma, Hep-G2 hepatocarcinoma, and A549 lung cancer cell lines [35]. Several other pieces of literature have reported a reduction in melanoma cell proliferation subsequent to treatment with antiretrovirals. In particular, a decrease in melanoma cell proliferation was observed after treatments with the anti-HIV protease inhibitors lopinavir [54] and nelfinavir [55]. Additionally, benzodithiazine-dioxide analogs exert both anti-proliferative and anti-HIV activity in a panel of cancer cell lines, including melanoma cells [56]. In these studies, the concomitant inhibition of HERV expression has not been evaluated. Despite the consistency of findings, including our data, suggesting a generalized inhibition of malignant cell growth with antiretroviral treatments, these results do not exclude the possibility of other mechanisms being involved in cell growth inhibition, distinct from HERV inhibition.

Metastasis poses a significant threat in melanoma, occurring frequently [2,3]. Once cells acquire an invasive phenotype, they can infiltrate other tissues, a process involving adhesion, extracellular matrix component proteolysis, and migration [57]. Migration is a prerequisite for invasion [57]. The assessment of malignant cell mobility was conducted through a wound healing assay. Despite inherent variations in mobility rates among different melanoma cell lines, cabotegravir consistently demonstrates its maximum efficacy in impeding melanoma cell mobility across all tested melanoma cell lines (Figure 3). Doravirine displayed significant effectiveness in A375 and MeWo cell lines, whereas lamivudine effectively inhibits cell mobility in MeWo and SK-Mel-28 cells (Figure 3). DTIC, in general, was less potent than cabotegravir in slowing down cell migration (Figure 3).

Measurements of anchorage-independent growth assess a cell’s ability to proliferate without the need of a solid surface for support, representing a distinctive feature of malignancy [58]. To explore the potential of antiretrovirals in inhibiting melanoma cell growth in an anchorage-independent context, a soft agar colony formation assay was conducted. Once more, cabotegravir demonstrated the ability to reduce both the number and size of colonies formed in soft agar in all melanoma cell lines (Figure 4). Lamivudine and doravirine exhibit limited effectiveness on SK-Mel-28 and MeWo cell lines; however, they demonstrate statistically significant inhibition of colony formation in A375 and FO-1 cell lines (Figure 4). Clonogenic ability has been associated with both the expression of HERVs and the outcomes of retroviral therapies in several studies. The use of RNA interference to downregulate HERV-K expression has been shown to inhibit the transition from adherent to non-adherent growth phenotypes [21]. Furthermore, cancer cells exhibiting stemness features demonstrate responsiveness to antiviral treatments, influencing both HERV transcription activity and clonogenic potential [35].

In summary, our findings, obtained from both BRAF-mutated and P53-mutated melanoma cell lines, suggest that two reverse transcriptase inhibitors belonging to the NRTI and NNRTI classes, along with a new-generation integrase inhibitor, impact HERV-K gene expression as well as the growth and invasion capabilities of melanoma cells.

Although cabotegravir is the latest FDA-approved antiviral drug for HIV treatment, clinical trials have demonstrated that patients have benefited from it with no prior instances of treatment failure and no resistance to treatment [59]. Its high efficacy at low doses and the high satisfaction reported by individuals living with HIV make it a very promising drug for long-term treatment [60]. Interestingly, some triazole analogues of cabotegravir were recently synthesized and tested in vitro for their antitumor activity against non-small-cell lung cancer cells [61] and a hepatocellular carcinoma cell line [62]. These results, along with our findings, suggest that the potential of cabotegravir in melanoma therapy deserves organoids and/or in vivo investigations.

Subsequently, through immunoblot assays, we explored the molecular mechanisms underlying the effects of lamivudine, doravirine, and cabotegravir on the malignant characteristics of melanoma cells. However, we limited the study to three BRAF-mutated cell lines—A375, FO1, and SK-Mel-28—in order to compare cell lines with more genotypical affinities.

STING serves as an initiator of innate immune signaling by functioning as a sensor for cytosolic DNA originating from bacteria and viruses. Thus, we expect that HERV-K expression may be positively correlated with the expression and activation of STING. The role of STING involves amplifying the production of interferon-stimulated genes and contributing to the host’s immune response [49]. Nevertheless, a link has been recently identified between a subclass of HERVs and the ability to trigger pathologic innate immune signaling in cancer involving STING and IFNγ, with important implications for PD-L1 expression and cancer immunotherapy [63]. It has been reported that inflammatory cytokines, particularly IFNγ, can upregulate the expression of PD-L1 in tumor cells, ultimately resulting in tumor immunosuppression [29,30]. We investigated the protein expression of the phosphorylated and activated forms of STING in melanoma cell lines after antiretroviral treatments. In the A375, FO-1, and SK-Mel-28 melanoma cell lines treated with lamivudine, doravirine, or cabotegravir, our immunoblots revealed a concurrent partial decrease in both STING phosphorylation and PD-L1 expression levels, while total STING protein expression remained unaffected (Figure 5). These findings may be attributed to the ability of antiretrovirals to inhibit the expression of HERV-K genes (Figure 2) and their subsequent replication. Notably, the decrease in the expression of the T-cell inhibitory molecule PD-L1 on the tumor cell surface has the potential to enhance effector T-cell function, thereby amplifying the immune response against the tumor [50]. Hence, in vitro treatment with these antiretroviral drugs appears to counteract the detrimental effects of PD-L1 expression activation through the STING/interferon pathway, thereby priming melanoma cells for an enhanced immune response.

Thereafter, our aim has been to investigate the molecular events implicated in the cytostatic and/or cytotoxic activity of antiretrovirals in melanoma cells.

Immunoblot findings reveal that antiretroviral treatment affects the viability of three BRAF-mutated cell lines through diverse mechanisms. Particularly in A375 cells, antiretroviral drugs—especially cabotegravir—demonstrate a cytostatic effect by inhibiting pRb expression levels. This effect is supported by a reduction in cyclin D1, as illustrated in Figure 6. A375 cells did not exhibit signs of apoptotic cell death. Nevertheless, a decrease in GPX4 and FTH1 levels, coupled with an upswing in transferrin expression, suggests the induction of ferroptosis, especially following cabotegravir treatment (Figure 7).

In the case of FO-1 cells, although no cytostatic effects are evident following each antiretroviral treatment, our immunoblot data suggest that both apoptosis and ferroptosis can be triggered, at least after cabotegravir administration (Figure 6 and Figure 7).

Concerning SK-Mel-28, the cytostatic and pro-apoptotic effects are only restricted to the cabotegravir treatment (Figure 6 and Figure 7).

Despite the absence of literature references on the regulatory effects of doravirine and cabotegravir on the cell cycle and cell death, a study conducted on a hepatocellular carcinoma cell line indicated that lamivudine was ineffective in reversing HBV-induced changes in cell cycle regulatory proteins [64]. Conversely, other studies have reported the pro-apoptotic effects of lamivudine in various cancer cell types, including hepatocellular carcinoma Huh-7 and Hep-G2 cell lines [65], as well as in breast [66], lung [67], and esophageal [68] cancer cells. It is worth noting that lamivudine is sometimes administered in conjunction with other therapeutic strategies, such as the chemotherapeutic agent paclitaxel or radiation.

Ferroptosis is a unique form of regulated cell death caused by lipid peroxidation in cellular membranes and is initiated by iron overload [3]. Over the last decade, accumulating evidence suggests that inducing ferroptosis could be a new, promising therapeutic approach for melanoma [3,69]. Some chemotherapeutic agents, but also radiotherapy and natural products, can induce ferroptosis in melanoma cells [3,70]. The induction of ferroptosis has been suggested to arise from either DNA damage or the release of reactive oxygen species [3,71]. As previously reported [72], raltegravir, an integrase inhibitor, was able to induce regulated cell death in multiple myeloma by damaging DNA. Consequently, cabotegravir could induce apoptosis and/or ferroptosis in melanoma cell lines, causing DNA damage. Cabotegravir, indeed, is capable of decreasing GPX4 and FTH1 in both A375 and FO-1 cell lines. It is also able to increase the cleaved form of PARP in FO-1 cells, which is a sign of apoptosis induction. Further investigations are needed to elucidate the molecular mechanism by which cabotegravir triggers apoptosis and ferroptosis in sensitive melanoma cells.

Ultimately, as indicated by the modest decrease in MMP2 expression levels in the BRAF-mutated cell lines (Figure 7), this finding may contribute to elucidating the previously reported reduction in cell mobility and colony formation within the same melanoma cell lines. 

Nevertheless, this study has some limitations. It is important to note that antiretroviral drugs may inhibit the expression of other HERV species besides HERV-K, which have not been investigated. Additionally, the study is conducted only in cell culture and lacks in vivo validation.

## 4. Materials and Methods

### 4.1. Cell Cultures

A375 (CRL-1619) and FO-1 (CRL-12177) melanoma cell lines and Normal Human Epithelial Melanocytes (NHEM, PCS-200-013) obtained from ATCC (Manassas, VA, USA) were cultured at 37 °C in a 5% CO_2_ humidified atmosphere. The culture medium consisted of high-glucose Dulbecco’s modified Eagle’s Medium (DMEM, Gibco, BRL Invitrogen Corp., Carlsbad, CA, USA), supplemented with 10% heat-inactivated fetal bovine serum (FBS; Gibco, BRL Invitrogen Corp., Carlsbad, CA, USA), and 1% antibiotic-antimycotic solution (Gibco, BRL Invitrogen Corp., Carlsbad, CA, USA).

SK-Mel-28 (HTB-72) and MeWo (HTB-65) melanoma cell lines obtained from ATCC (Manassas, VA, USA) were cultured in Roswell Park Memorial Institute 1640 medium (RPMI-1640, Gibco, BRL Invitrogen Corp., Carlsbad, CA, USA) under the same aforementioned conditions.

### 4.2. RNA Extraction, Reverse Transcription, and Real-Time PCR

SK-Mel-28, A375, FO-1, and MeWo cell lines were seeded in 6 mm petri dishes (SK-Mel-28: 250 × 10^3^ cells/petri; A375: 150 × 10^3^ cells/petri; FO-1: 200 × 10^3^ cells/petri; MeWo: 300 × 10^3^ cells/petri). At 80% confluence, all the cell lines were treated with 1 µM lamivudine, 5 µM doravirine, and 3 µM cabotegravir (MERK, Milan, Italy). After 24 h of treatment, total RNA for gene expression analyses was extracted, following the manufacturer’s protocol, with Trizol Reagent (ThermoFisher Scientific, Milan, Italy). RNA quantification was performed with a Tecan NanoQuant Infinite M200 Pro plate reader (Tecan Group Ltd., Männedorf, Switzerland), and its quality was checked through 1% Agarose gel electrophoresis. RNA (500 ng) was reverse transcribed by the SensiFAST cDNA Synthesis Kit (Bioline, Trento, Italy), following the manufacturer’s protocol. After reverse transcription, HERV-K gene (Pol, Env) expression levels were determined by real-time polymerase chain reaction (RT-PCR). Normalization was performed using the cluster of differentiation 151 protein (CD151), which turned out to be the most stable gene in our experimental conditions.

The amplification primers were the following:
CD151 (Fw) 5′-CTACGCCTACTACCAGCAGC-3′, 
(Rv) 5′-CGGAACCACTCACTGTCTCG-3′Pol (Fw) 5′-CCACTGTAGAGCCTCCTAAACCC-3′, 
(Rv) 5′-GCTGGTATAGTAAAGGCAAATTTTTC-3′Env (Fw) 5′-GCCATCCACCAAGAAAGCA-3′, 
(Rv) 5′-AACTGCGTCAGCTCTTTAGTTGT-3′

Real-time PCR was carried out with the Bio-Rad CFX Connect Real-Time System using the SensiFAST SYBR No-ROX Kit (Bioline, Trento, Italy). The following amplification protocol was used: 2 min step at 95 °C for polymerase activation, followed by 40 cycles of 5 sec denaturation at 95 °C and a 20 sec primer annealing polymerization step at 60 °C. Each measurement was carried out in triplicate for at least four different experiments.

### 4.3. Cell Viability Assay

Melanoma cells were plated in 96-well plates (SK-Mel-28: 4.0 × 10^3^ cells/well; A375: 2.9 × 10^3^ cells/well; FO-1: 3.5 × 10^3^ cells/well; MeWo: 6 × 10^3^ cells/well; NHEM: 6 × 10^3^ cells/well). The following day, the cells underwent treatment with antiretroviral drugs or the antineoplastic alkylating agent DTIC (MERK, Milan, Italy) with an incubation period of 72 h. Upon completion of the treatment, cell fixation was achieved by adding 25 µL/well of 50% (*w*/*v*) trichloroacetic acid to the culture medium. Subsequently, the plates were incubated at 4 °C for 1 h, washed four times with deionized water (ddH2O), and left to air-dry at room temperature (RT).

For staining, 50 µL/well of 0.04% (*w*/*v*) sulforhodamine B (SRB) sodium salt solution (Sigma-Aldrich, Milan, Italy) was applied. After a 1-h incubation at RT, the plates were rinsed with 1% acetic acid and allowed to air-dry. The SRB stain was then solubilized using a 10 mM Tris-base solution at pH 10.5. Subsequently, the absorbance of the samples was measured at 540 nm using a TECAN NanoQuant Infinite M200 Pro plate reader (Tecan Group Ltd., Männedorf, Switzerland). Each condition or data point was subjected to four to eight replicates.

### 4.4. Wound-Closure Cell Migration Assay

A375, FO-1, SK-Mel-28, and MeWo cell migration were assessed using the wound-closure cell migration assay, commonly known as a scratch test. Cells were seeded in a 12-well plate (A375, FO-1: 150 × 10^3^ cells/well; SK-Mel-28, MeWo: 200 × 10^3^ cells/well). Upon reaching confluence, the wells were washed with DPBS. Subsequently, the monolayers were scratched using a sterile pipette tip. To eliminate detached cells, wells were washed with complete medium and replenished with DPBS 1X or complete medium with or without treatment. The cells were then incubated for 48 h at 37 °C in a humidified atmosphere with 5% CO_2_.

Images of cell movement were captured at different times using an inverted microscope (Axio Vert A1, Zeiss, Oberkochen, Germany). The acquired images were quantitatively analyzed using ImageJ computing software with the MRI wound healing tool.

### 4.5. Colony Formation Assay in Soft Agar

The anchorage-independent growth of A375, FO-1, SK-Mel-28, and MeWo melanoma cells was assessed through colony formation in soft agar, as previously described [73]. Initially, the bottom layer of 6-well plates was filled with 1% low gelling temperature agarose (Sigma-Merck, Milan, Italy) dissolved in 2X DMEM, 20% FBS, and 2% antibiotic-antimycotic solution. Subsequently, 0.6% low gelling temperature agarose, also dissolved in 2X DMEM, 20% FBS, and 2% antibiotic-antimycotic solution, was layered over the 1% agarose, along with treated or untreated cells (A375, FO-1: 7.0 × 10^3^ cells/well, and SK-Mel-28, MeWo: 16 × 10^3^ cells/well).

Fresh media (200 μL) was added to each well twice a week. After 15–21 days, the formed cell colonies were observed under an inverted microscope (Axio Vert A1, Zeiss, Oberkochen, Germany).

### 4.6. Total Protein Extracts

Cells were seeded in 35 mm petri dishes (SK-Mel-28: 100 × 10^3^ cells/petri; A375: 80 × 10^3^ cells/petri; FO-1: 90 × 10^3^ cells/petri; MeWo: 140 × 10^3^ cells/petri). After 24 h, the cells underwent treatment with two different concentrations of each drug or were left untreated as a control. Following an additional 24 or 48 h of treatment, the cells were scraped using warm 1× sample buffer (2% SDS, 10% glycerol, 50 mM Tris-HCl, 1.75% β-mercaptoethanol, and bromophenol blue) and subsequently boiled at 98 °C for 10 min. The total protein extracts were then preserved at −80 °C until further analysis.

### 4.7. Immunoblot Analysis

Protein extracts were subjected to electrophoresis using an 8.5–12% polyacrylamide SDS-PAGE gel. Proteins were subsequently transferred onto a polyvinylidene difluoride membrane (PVDF, Merck-Millipore, Milan, Italy). The membranes were then blocked at RT using a TBST buffer (10 mM Tris-HCl pH 7.5, 100 mM NaCl, 0.1% Tween 20) containing 5% milk for 1 h. Following this, membranes were incubated overnight at 4 °C on a shaker using a 5% BSA solution containing the primary antibody against Transferrin (A1448), Ferritin heavy chain (A19544), pRb Ser807/811 (AP0484), STING (A21051), pSTING (AP1369), cleaved PARP (A19612), PD-L1 (A1645) (Abclonal, Woburn, MA, USA); Cyclin D1 (GTX634347), MMP-2 (GTX634832) (Genetex, Alton Parkway, Irvine, CA, USA); GPX-4 (67763-1), and α-tubulin (66031-1) (Proteintech, Manchester, UK). Subsequently, the membranes were washed three times for 10 min each with TBST buffer. They were then incubated for 1 h with a horseradish peroxidase-conjugated secondary antibody, either anti-rabbit or anti-mouse (Cell Signaling Technology, Danvers, MA, USA). After this incubation, membranes were washed again 3 times for 10 min each with TBST buffer. The expression level of each protein was normalized using α-tubulin protein levels, unless otherwise specified. Immuno-detection was performed using an ECL kit (Merck-Millipore, Milan, Italy), and the chemiluminescence signals were visualized using ChemiDoc (Bio-Rad, Hercules, CA, USA) equipment.

### 4.8. Statistics

The results are presented as the mean value ± standard deviation (S.D.). Statistical differences were analyzed using the GraphPad Prism statistical program, employing an unpaired, two-tailed Student’s t-test, unless otherwise stated. A *p*-value less than 0.05 (*) or less than 0.01 (**) was considered to be statistically significant. Each type of experiment was conducted with a minimum of three independent biological replicates. The normal distribution of the data was assessed using the Shapiro–Wilk test.

## 5. Conclusions

Cutaneous melanoma, being a highly aggressive tumor type, remains challenging to eradicate completely with the currently available therapeutic strategies [2,3]. Notably, HERVs exhibit upregulation across various tumor types, with a particularly pronounced presence in melanoma [17,18,19]. Our investigation demonstrates that antiretrovirals belonging to the NRTI, NNRTI, and INSTI classes can downregulate the expression of HERV-K genes in four distinct melanoma cell lines, thereby limiting the aggressive potential of these cells. Specifically, cabotegravir, a novel antiretroviral drug targeting integrase, proves to be a potent tool for inhibiting cell growth and invasion potential through various mechanisms in different cell lines. Beyond its antiretroviral effects, cabotegravir holds promise for potential use in conjunction with other chemotherapeutic agents, as it exhibits the ability to induce cell growth inhibition, apoptosis, ferroptosis, and hinder migration and adhesion of melanoma cells.

Lastly, given the significance of the PD-1/PD-L1 pathway as a crucial target in cancer immunotherapy, chemotherapeutic molecules with the ability to downregulate PD-L1 expression in tumor cells hold promise for enhancing the immune response. The retroviral drugs examined in this in vitro study demonstrated the capacity to inhibit the expression of PD-L1 antigen in melanoma cells. These findings also raise the prospect of using antiretroviral drugs to downregulate PD-L1 expression, potentially enhancing the therapeutic responses of immune checkpoint inhibitors.

## Figures and Tables

**Figure 1 ijms-25-01615-f001:**
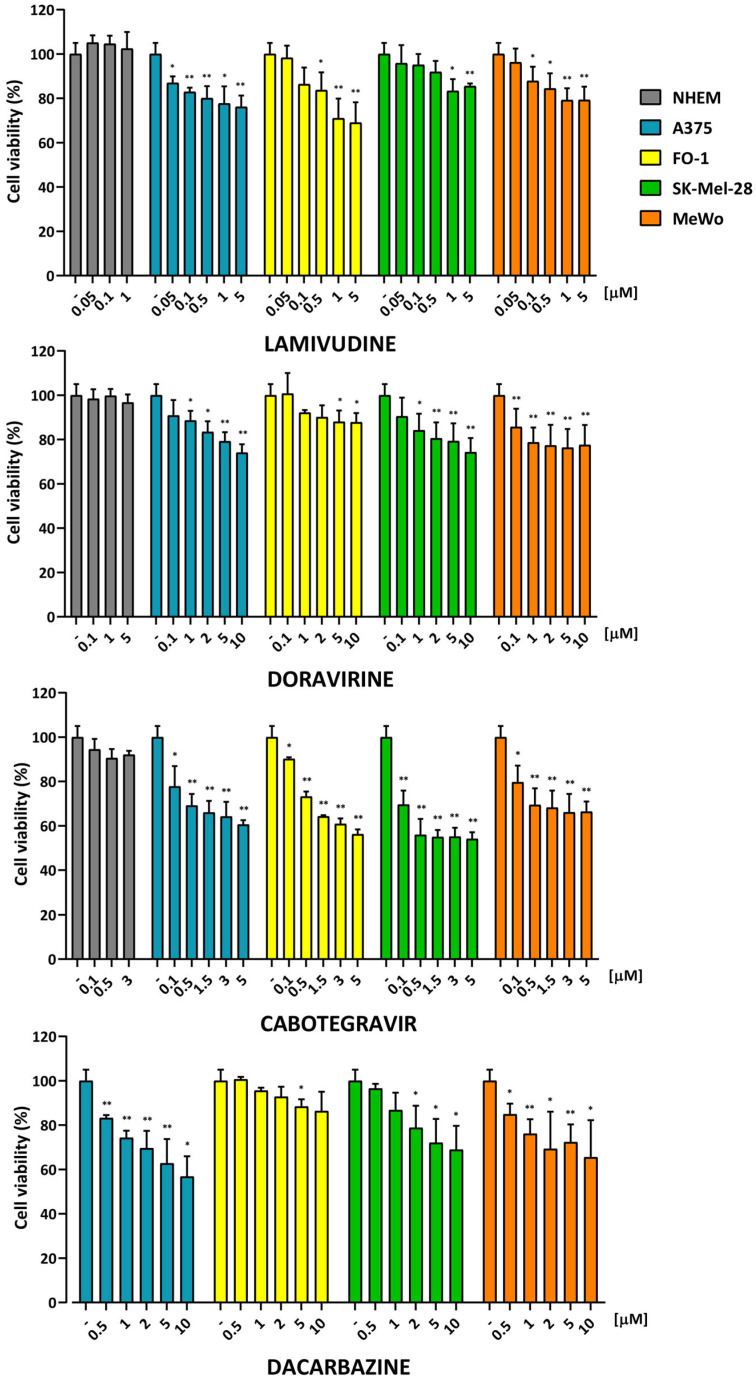
Concentration-dependent reduction of cell viability after antiretroviral treatments. Cell viability was assessed using the Sulforhodamine B (SRB) assay after 72 h of treatment with antiretroviral drugs in four human melanoma cell lines (SK-Mel-28, A375, FO-1, and MeWo), as well as in normal human epithelial melanocytes (NHEM). Dacarbazine, a medication commonly used in the treatment of melanoma, was employed for comparison. Data were acquired by calculating the mean ± S.D. of values from at least four independent experiments, which were then compared with the untreated control; * *p* < 0.05; ** *p* < 0.01.

**Figure 2 ijms-25-01615-f002:**
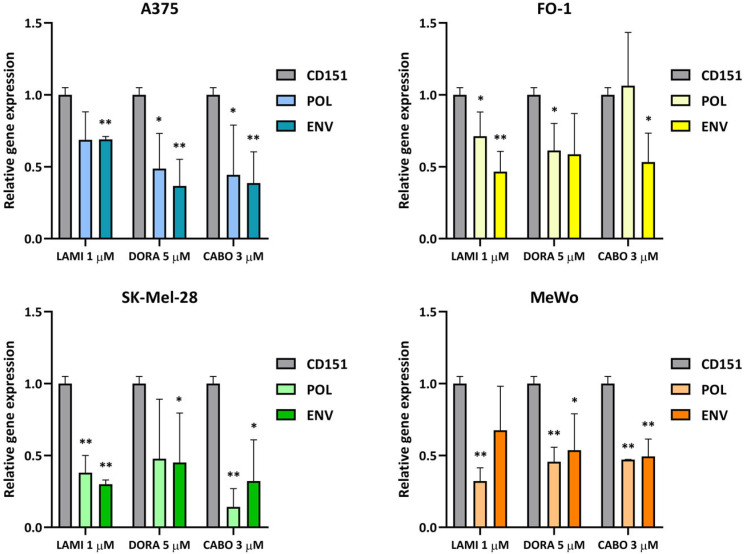
Relative expression of HERV-K Pol and Env genes 24 h after each antiretroviral treatment. Cells were treated with lamivudine (1 µM), doravirine (5 µM), or cabotegravir (3 µM) for 24 h (abbreviated as LAMI, DORA, and CABO, respectively). The mRNA expression levels of treated samples were measured by RT-PCR and compared with those of untreated melanoma cells. The bars represent the mean values ± standard deviation (S.D.) of four independent experiments. All comparisons were made against each control sample after normalization with CD151 expression; * *p* < 0.05; ** *p* < 0.01.

**Figure 3 ijms-25-01615-f003:**
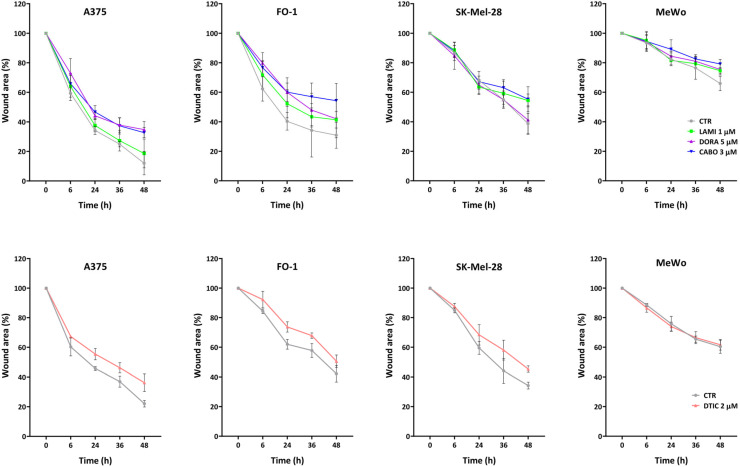
Wound healing assay of melanoma cell lines treated with antiretroviral drugs. The time lapse of the wound healing assay confirms the inhibition of melanoma cell mobility following treatment with lamivudine (1 µM), doravirine (5 µM), or cabotegravir (3 µM) (abbreviated as LAMI, DORA, and CABO, respectively; CTR, untreated cells). Dacarbazine (DTIC), a medication commonly used in the treatment of melanoma, was employed for comparison. Images were captured using an inverted microscope (Axio Vert A1, Zeiss, Oberkochen, Germany), and quantitative analysis was performed using ImageJ computing software (Version: 2.14.0/1.54f. NIH Image, Bethesda, MD, USA). The statistical significance is reported in the Results section.

**Figure 4 ijms-25-01615-f004:**
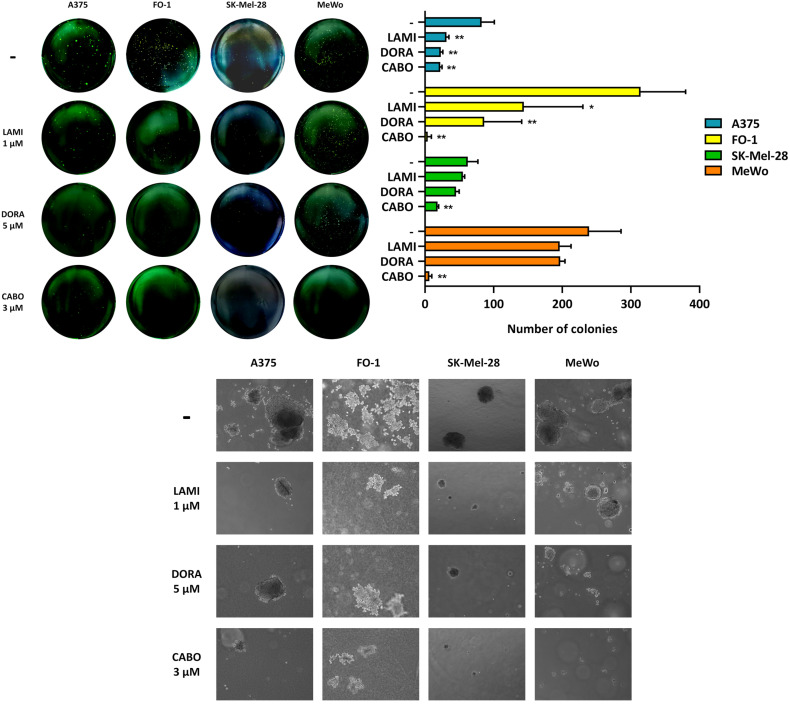
Antiretroviral drugs inhibit soft agar colony formation in melanoma cell lines. Images of colonies are representative examples taken from experiments illustrating colony formation in soft agar. The histograms represent the quantification of colony number for each antiretroviral treatment, while the microscopic images depict the size and density of colonies (5× inverted microscopy, Axio Vert A1, Zeiss, Oberkochen, Germany). All comparisons were performed vs. each control sample; * *p* < 0.05; ** *p* < 0.01.

**Figure 5 ijms-25-01615-f005:**
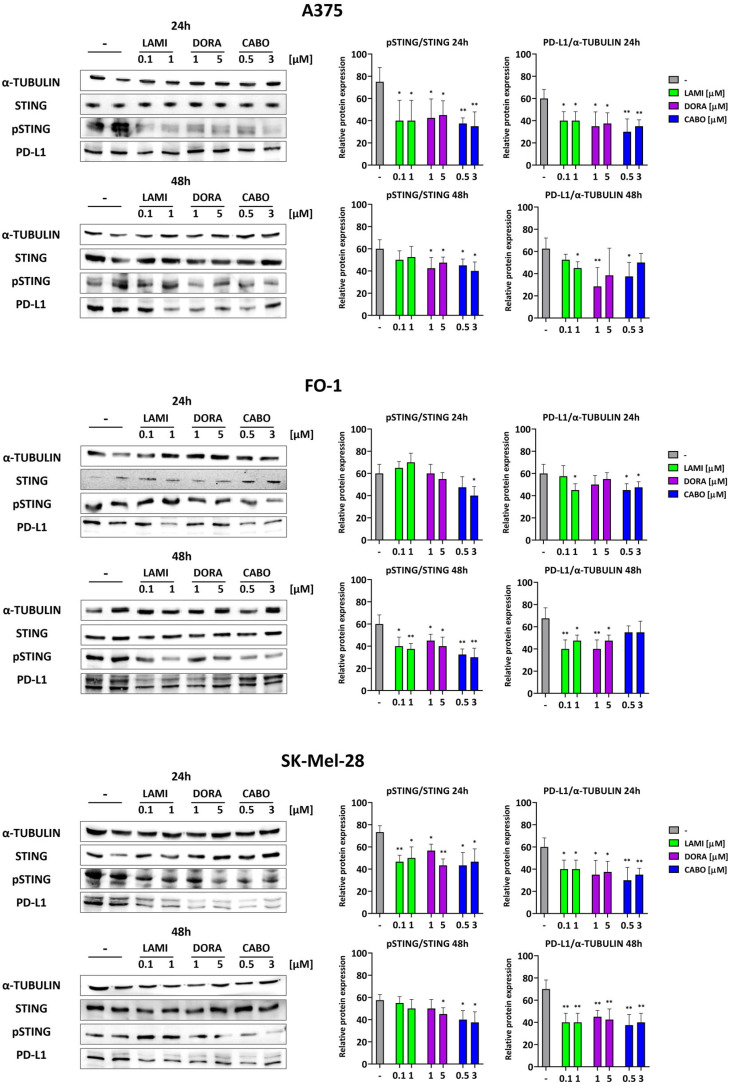
The expression levels of proteins associated with the immune response are impacted by antiretroviral drugs. Representative immunoblots are shown on the left. The phosphorylated and activated form of the stimulator of interferon genes (STING) and the protein expression levels of programmed death-ligand 1 (PD-L1) decreased at 24 and/or 48 h following antiviral treatments. On the right, histograms represent the mean values ± S.D. of protein expression level measured by densitometry derived from three independent experiments and normalized with α-tubulin expression, whereas pSTING results were normalized with total STING protein level. All comparisons were performed vs. each control sample after data normalization; * *p* < 0.05; ** *p* < 0.01.

**Figure 6 ijms-25-01615-f006:**
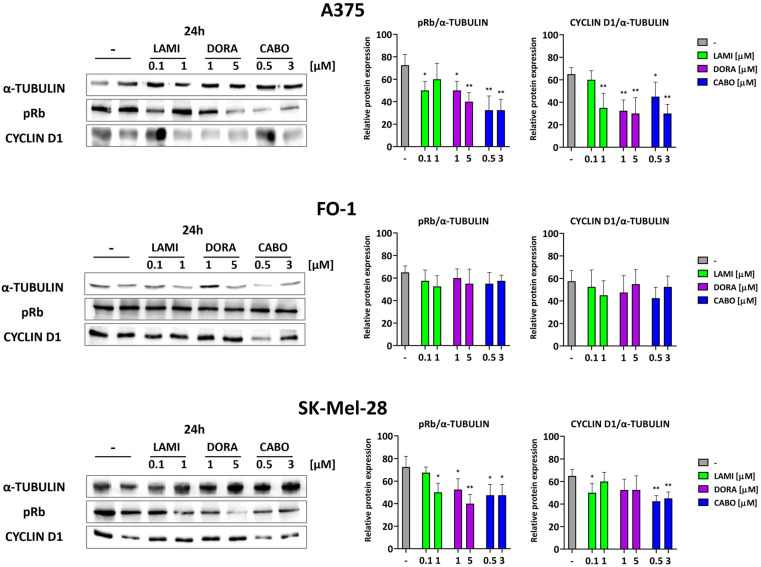
Representative immunoblots show the effects of antiretrovirals on the expression of proteins regulating the cell cycle. Antiretroviral treatments downregulate cell cycle regulatory proteins in A375 and SK-Mel-28, but not in the FO-1 cell line. pRb (phosphorylated form of retinoblastoma protein). Histograms represent the mean values ± S.D. of protein expression level measured by densitometry derived from three independent experiments and normalized with α-tubulin expression. All comparisons were performed vs. each control sample after data normalization; * *p* < 0.05; ** *p* < 0.01.

**Figure 7 ijms-25-01615-f007:**
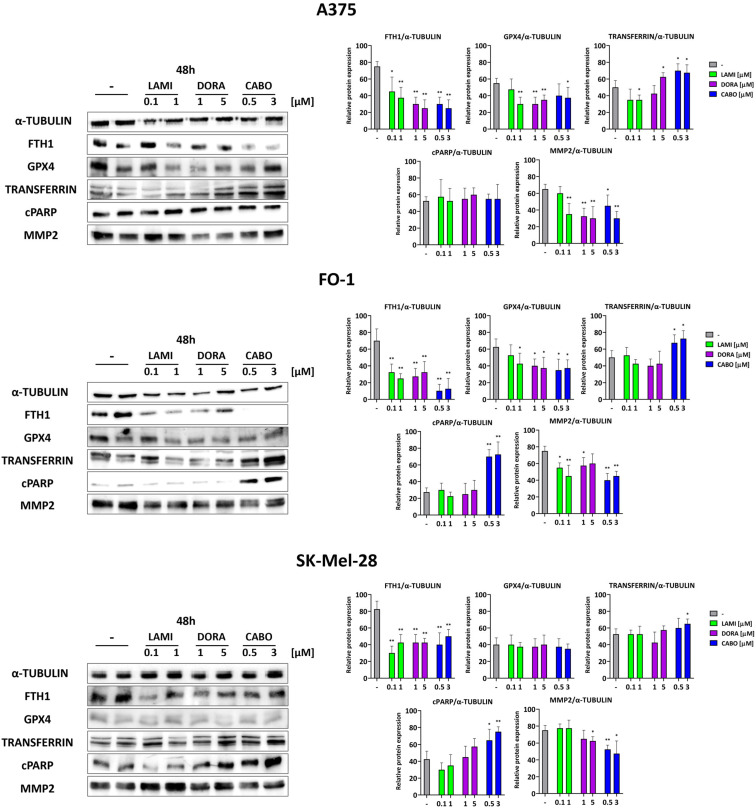
Representative immunoblots show the effects of antiretrovirals on the expression of proteins regulating cell death. The antiretroviral cabotegravir induced apoptosis in SK-Mel-28 and FO-1 melanoma cells but not in the A375 melanoma cell line, as suggested by the high expression of the cleaved form of poly (ADP-ribose) polymerase (cPARP). Ferroptosis is induced in all cell lines, as suggested by a decrease in ferritin (FTH1) and/or glutathione peroxidase 4 (GPX4) expression and increased transferrin. Metalloproteinase 2 (MMP2) protein expression is sharply decreased in A375 FO-1, and SK-Mel-28 cell lines. Histograms represent the mean values ± S.D. of protein expression level measured by densitometry derived from three independent experiments and normalized with α-tubulin expression. All comparisons were performed vs. each control sample after data normalization; * *p* < 0.05; ** *p* < 0.01.

## Data Availability

Data are contained within the article.

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
