# Peer review of "Lamivudine, Doravirine, and Cabotegravir Downregulate the Expression of Human Endogenous Retroviruses (HERVs), Inhibit Cell Growth, and Reduce Invasive Capability in Melanoma Cell Lines"

_ijms, 2024, doi:10.3390/ijms25031615_

Round 1
Reviewer 1 Report
Comments and Suggestions for Authors
The study aims to test the well-known antiretroviral drugs lamivudine, doravirine and cabotegravirna as potential antitumor agents against melanoma cell lines containing mutations. The authors identify their effect on the expression of human endogenous retroviruses (HERVs) as the main mechanism of antitumor action of these antiretroviral drugs. Taking into account the extreme prevalence of melanoma, as well as its high malignancy and ability to metastasize, the potential for the search for new agents for the treatment of melanoma is high. The work may be of interest, but its design requires revision:
1) In experiments assessing the cytotoxicity of the tested drugs against NHEM and various melanoma cell lines, it is necessary to provide the values of cell survival in the absence of the active substance (for control), as well as in comparison with any standard cytostatic preparticle used for the treatment of this type of tumor. These data are necessary to evaluate the efficacy of the approach proposed by the authors for the treatment of cancer.
2) In Figure 1 for lamivudine, the values obtained, according to the histogram, exceed 100% survival. The authors should revise and correct the figure.
3) Authors should improve the quality of figures, captions are not readable and too small, the design of figures does not meet the high quartile of IJMS journal.
4) Authors should correct the subheadings of sections 2.3. and 2.4.
5) The authors should decode the abbreviation CTR in Figure 2. The authors should also provide data on melanoma cell migration ability both in the absence of the tested antiretroviral preparations and in the presence of a standard cytostatic drug.
6) The results presented in section 2.4 should be shown in the diagram, as it is difficult for the reader to visually evaluate the degree of colony formation. It is also necessary to compare the data obtained with the data for control samples without the antiretroviral preparative being tested and in the presence of the standard cytostatic drug.
7) Figure 5, 6 needs careful revision. The captions are not legible. The quality of presentation of the graphical material does not correspond neither to the high level of the IJMS journal nor to the interesting results obtained. Data for control groups should be added.
8) The beginning of the Discussion section repeats the material in the Introduction section and does not contribute to the discussion of the results obtained by the authors.
9) The authors should add more information in the introduction about currently available examples of successful use of antiretroviral drugs for tumor treatment.
Author Response
Reply to Reviewer1:
We appreciate the reviewer's positive feedback and for the valuable suggestions aimed at improving our work.
Responses to the Reviewer's Criticisms
- In experiments assessing the cytotoxicity of the tested drugs against NHEM and various melanoma cell lines, it is necessary to provide the values of cell survival in the absence of the active substance (for control), as well as in comparison with any standard cytostatic preparticle used for the treatment of this type of tumor. These data are necessary to evaluate the efficacy of the approach proposed by the authors for the treatment of cancer.
Authors’ Reply:
Thanks for the suggestions. We implemented our cell viability evaluation with a chemotherapy drug that has been used in clinic against melanoma. We excluded the most effective BRAF inhibitors because MeWo cell line has a BRAF wild type genotype. We chose dacarbazine (DTIC), a methylating agent used in primary and metastatic melanoma therapy. Cell viability findings are showed in the new Figure 1.
- In Figure 1 for lamivudine, the values obtained, according to the histogram, exceed 100% survival. The authors should revise and correct the figure.
Authors’ Reply:
Only the histograms reporting cell viability findings with 0.05 and 0.1 µM lamivudine treatment in NHEM cells slightly exceeded the mean value of untreated control samples. This observation may suggest a pro-proliferative effect of lamivudine in these normal cells. However, the comparison with untreated control data was not statistically significant, and, in our opinion, it might simply reflect normal experimental variability. Thus, the findings shown are not a graphic error but real results.
- Authors should improve the quality of figures, captions are not readable and too small, the design of figures does not meet the high quartile of IJMS journal.
Authors’ Reply:
We concur with the reviewer; all figures have been restyled for improved quality and readability.
- Authors should correct the subheadings of sections 2.3. and 2.4.
Authors’ Reply:
Titles of subsections 2.3 and 2.4 have been modified, as suggested.
- The authors should decode the abbreviation CTR in Figure 2. The authors should also provide data on melanoma cell migration ability both in the absence of the tested antiretroviral preparations and in the presence of a standard cytostatic drug.
Thanks for the suggestions. We implemented our cell mobility evaluation with a chemotherapy drug (DTIC) that has been used in clinic against melanoma. The findings are shown in the new Figure 3.
- The results presented in section 2.4 should be shown in the diagram, as it is difficult for the reader to visually evaluate the degree of colony formation. It is also necessary to compare the data obtained with the data for control samples without the antiretroviral preparative being tested and in the presence of the standard cytostatic drug.
Thank you for the suggestions. For each cell line, we quantified the number of colonies in soft agar for the untreated control and all antiretroviral treatments. Histograms and statistical analyses have been inserted into the new Figure 4. Unfortunately, due to the extended duration required for each experiment (3-4 weeks for a single experiment), we were unable to test the effects of DTIC as a positive control.
- Figure 5, 6 needs careful revision. The captions are not legible. The quality of presentation of the graphical material does not correspond neither to the high level of the IJMS journal nor to the interesting results obtained. Data for control groups should be added.
We concur with the reviewer; the figures 5 and 6 have been restyled for improved readability.
- The beginning of the Discussion section repeats the material in the Introduction section and does not contribute to the discussion of the results obtained by the authors.
Thank you for the suggestions. As advised, we delated two sentences at the beginning of the Discussion Section.
- The authors should add more information in the introduction about currently available examples of successful use of antiretroviral drugs for tumor treatment.
We agree with the reviewer. As also recommended by another reviewer, we have enhanced the Introduction Section with additional data concerning the antitumor effects of antiretrovirals. The following description has been added: (lines 134-151 of the revised version)
“To the best of our knowledge, antiviral drugs have not been approved as anti-proliferative agents. However, promising in vitro studies have demonstrated that reverse transcriptase inhibitors can modulate cell growth and differentiation across various cancer types, as reviewed by [31]. For instance, amantadine, ribavirin and pleconaril, tested against HCT8 colonic cancer cells, were shown to have cytotoxic activity and to force the downregulation of HERV proteins [32]. Moreover, ritonavir, atazanavir, and lopinavir have been shown to reduce the proliferation of schwannoma and grade I meningioma cells [33]. Additionally, nevirapine, has demonstrated the ability to reduce cell growth and promote differentiation in acute myeloid leukemia (AML) cells and primary blasts from two AML patients [12]. Landriscina et al. reported reversible inhibition of cell proliferation in an undifferentiated thyroid carcinoma cell line with high endogenous RT activity upon treatment with the NNRTIs nevirapine and efavirenz [34]. These NNRTIs were also tested on CD133-positive melanoma stem cells, showing a downregulation of HERV-K activity concurrent with a decrease of the proliferative rate and the CD133 gene expression [9]. Recently, Giovinazzo et al. compared the effects of the NRTI azidothymidine and the NNRTI efavirenz on cell lines derived from lung adenocarcinoma, hepatocellular carcinoma, and melanoma [35]. The treatments affected HERVs’ transcriptional activity in parallel with the reduction of cell growth, clonogenic activity, and induction of apoptosis [35].”
Reviewer 2 Report
Comments and Suggestions for Authors
The study aims to demonstrate the effectiveness of three antiretroviral drugs against cultured human melanoma cells, analyzing some different aspects, all related to the malignancy of the disease, investigating also the molecular mechanisms underlying the observed effects.
In consideration of the severity of the disease and the constant increase in its incidence, the evaluation of the effectiveness of drugs, which are already used in the clinical practice for other pathologies, represents a theme of particular interest.
The choice of cell lines used is well motivated as well as the procedure followed to determine the concentrations of the drug to be used is correct.
The study is well structured and the methodologies adopted are adequate and well explained.
The description of the results is clear and sufficiently detailed. The discussion takes into consideration all the essential points and refers to the most up-to-date literature.
Overall, the manuscript provides a valid contribution to knowledge relating to the therapy of melanoma, highlighting the effectiveness of reverse transcriptase and integrase inhibitors on what are now known as the main characteristics of the cellular component of this important neoplasm.
I agree with the authors on the limitation of being an in vitro study but I underline that the in vitro methodology must always come before preclinical studies, also in considerations of ethical reasons.
As regards the extension of the study to other HERV families, I believe it would be desirable, in consideration of the recent studies that suggest their involvement also in human melanoma. In this regard perhaps references of the most recent literature could be added.
I confidentially suggest that the authors indicate HERV-K as a family within human endogenous retroviruses and not rather as a species.
Minor revision.
The authors in the discussion section wrote: "We explored the capacity of two RT inhibitors: the nucleoside analogue RT ................ as well as the integrase inhibitor, to suppress HERV-K Pol and Env gene expression"
The existing literature not demonstrated a "suppression" of expression but rather a downregulation of the expression. Therefore, I suggest to modify the statement.
In several parts of the manuscript the authors write: "to assess the anti-melanoma activities of....". Although I believe the statement to be correct, if by melanoma we mean cells derived from melanoma, in the current meaning melanoma is the name of an oncological skin disease. So I would eliminate "anti-melanoma"
In the discussion section, referring to the colony-forming ability of melanoma cell lines treated in vitro with antiretroviral drugs, the authors state "In the case of SK-Mel-28, only a few colonies were observed in the treated cells, indicating differences limited between treated and control samples (Figure 4)". I disagree with the logic of this statement. The event to be considered is that the control cells form few colonies and not that the treated ones are quantitatively similar, as was hypothesized.
​Line 69. I propose to add that HERV-K is among the endogenous retroviruses the one that has most recently integrated into the human genome and therefore has been able to preserve, more than others, its ancestral characteristics. In fact, it is the family most frequently associated with human pathologies
Line 253. “In the case of SK-Mel-28, the diminished formation of colonies in both control and treated samples ….” The meaning of this statement is unclear. Decreased colony formation compared to what, since we're talking about control? The term “diminished” is incorrect
Figure 5 should be made more easily readable
Line 332. vs should be changed to vs
Line 364. in vitro should be changed to in vitro. "In vitro" must be written in italics throughout
Line 369. In the Conclusions section the authors state: "investigation demonstrates that antivirals employing three different mechanisms of action can suppress HERV-K expression............."
The three drugs used act against the virus in 2 distinct ways, two being (Lamivudine and Doravirine) RT inhibitors and one (Cabotegravir) integrase inhibitor. What did the authors mean? Please explain
Line 472. Replace figures with Figures
C
Author Response
Reply to Reviewer:
We appreciate the reviewer's positive feedback and thank him for valuable suggestions aimed at improving the manuscript's readability
Responses to the Reviewer's Criticisms
- As regards the extension of the study to other HERV families, I believe it would be desirable, in consideration of the recent studies that suggest their involvement also in human melanoma. In this regard perhaps references of the most recent literature could be added. I confidentially suggest that the authors indicate HERV-K as a family within human endogenous retroviruses and not rather as a species.
Reply to Reviewer:
We agree with the Reviewer. We replaced ‘family’ instead of ‘species’.
- The authors in the discussion section wrote: "We explored the capacity of two RT inhibitors: the nucleoside analogue RT ................as well as the integrase inhibitor, to suppress HERV-K Pol and Env gene expression" The existing literature not demonstrated a ‘suppression’ of expression but rather a downregulation of the expression. Therefore, I suggest to modify the statement.
Reply to Reviewer:
We agree with the Reviewer’s suggestion. We replaced ‘suppression’ with ‘downregulation’ in both Abstract, Introduction, and Discussion Sections.
- In several parts of the manuscript the authors write: "to assess the anti-melanoma activities of....". Although I believe the statement to be correct, if by melanoma we mean cells derived from melanoma, in the current meaning melanoma is the name of an oncological skin disease. So I would eliminate "anti-melanoma"
Reply to Reviewer:
We agree with the Reviewer’s suggestion. In one part we delated ‘anti-melanoma’. In the others, we replaced with “To assess the impact of lamivudine, doravirine, and cabotegravir on melanoma cell growth” or “Subsequently, we explored molecular mechanisms underlying the effects of lamivudine, doravirine, and cabotegravir on the malignant characteristics of melanoma cells.”
- In the discussion section, referring to the colony-forming ability of melanoma cell lines treated in vitro with antiretroviral drugs, the authors state "In the case of SK-Mel-28, only a few colonies were observed in the treated cells, indicating differences limited between treated and control samples (Figure 4)". I disagree with the logic of this statement. The event to be considered is that the control cells form few colonies and not that the treated ones are quantitatively similar, as was hypothesized.
Reply to Reviewer:
We agree with the Reviewer’s criticism.
We quantified the colony numbers for all treatments and illustrated the data in a diagram (New Figure 4). Following the new assessments and the statistical analysis, the sentence has been amended as follows:
“Figure 4 presents both the quantification of colony number and representative pictures of colony size and density for each antiretroviral treatment and the untreated control. Despite lamivudine and doravirine demonstrating proficient efficacy in reducing the number of colonies in A375 and FO-1, their impact on SK-Mel-28 and MeWo cell lines appears relatively modest, with values not reaching statistical significance (Figure 4). Instead, cabotegravir was found to be the most effective treatment, leading to a massive reduction in both colony size and number in A375 and FO-1 cell lines, and preventing colony formation in SK-Mel-28 and MeWo cell lines (Figure 4).”
- Figure 5 should be made more easily readable
Reply to Reviewer:
Figure 5 and 6 have been modified and enlarged to make them more readable.
- Line 332. vs should be changed to vs
Reply to Reviewer:
Done.
- Line 364. in vitro should be changed to in vitro. "In vitro" must be written in italics throughout.
Reply to Reviewer:
All instances of 'in vitro' in the text have been changed to italics
- Line 369. In the Conclusions section the author’s state: "investigation demonstrates that antivirals employing three different mechanisms of action can suppress HERV-K expression............."
The three drugs used act against the virus in 2 distinct ways, two being (Lamivudine and Doravirine) RT inhibitors and one (Cabotegravir) integrase inhibitor. What did the authors mean? Please explain
Reply to Reviewer:
Thank you for the suggestion. The sentence has been changed as follows: “Our investigation demonstrates that antiretrovirals belonging to the NRTIs, NNRTIs, and INSTIs classes can downregulate the expression of HERV-K genes in four distinct melanoma cell lines, thereby limiting the aggressive potential of these cells.”
- Line 472. Replace figures with Figures
Reply to Reviewer:
Done
Reviewer 3 Report
Comments and Suggestions for Authors
Authors explores the impact of the antiretrovirals administration on the expression of human endogenous retroviruses (HERVs), cell growth and invasive capability of human melanoma cell lines in culture. They investigated three antiretrovirals—lamivudine, doravirine, and cabotegravir—in A375, FO-1, and SK-Mel-28, BRAF-mutated and in MeWo, P53-mutated, melanoma cell lines. The findings indicate a general capability of these drugs to suppress the expression of HERV K Pol and Env genes and hinder cell viability, mobility, and colony formation capacity of melanoma cells. This is interesting topic and this study can be accepted after minor revision.
Here are the points:
· Cancer is a common comorbidity with HIV/AIDS. Authors should mention about it. In particular what is incidence of melanoma with HIV/AIDS?
· Authors chose Lamivudine, Doravirine, and Cabotegravir, two revers transcriptase and one integrase inhibitor. Authors should also explain integrase inhibitors in Intoduction that they only explained revers transcriptase inhibitors. Even they should explain the difference of new non-nucleoside reverse transcriptase inhibitors or nucleoside reverse transcriptase inhibitors due to difference of lamivudine and doravirine.
· Authors should explain the potential of combinations of these drugs as they generally are used in combinations
· What about their drawbacks in pharmacokinetic profiles? Is it safe to use these drugs?
· Why they choose these three drugs? According to chemical structure, safety,,,,? What are their superiorities?
· In experiments, authors barely use quantitative data in Results particularly? Authors should compare the results based on the calculation results to be more scientific.
· Authors indicate that drugs to suppress the expression of HERV K Pol and Env genes. What about protease inhibition as it is included in Pol genes?
Author Response
Reply to Reviewer3:
We appreciate the reviewer's positive feedback and for the valuable suggestions aimed at improving the manuscript's readability.
Responses to the Reviewer's Criticisms
- Cancer is a common comorbidity with HIV/AIDS. Authors should mention about it. In particular what is incidence of melanoma with HIV/AIDS?
Authors’ Reply:
Thanks for the suggestions. We mentioned that cancer is a common comorbidity with HIV as suggested. We added in the Discussion Section the followed sentences:
“It is important to note that individuals with HIV/AIDS require long-term support through the Highly Active Antiretroviral Therapy (HAART), which may influence the occurrence of associated malignancies. The documented incidence of melanoma among people living with HIV/AIDS varies from 16.4 to 175.7 per 100,000 person-years in selected American and European cohorts during the early HAART era [51]. Until the first decade of the current century, most studies, but one, indicated no discernible cumulative trend in melanoma incidence among individuals living with HIV/AIDS compared to the general population [51]. Conversely, there is a notable increase in the risk of melanoma among other immunosuppressed groups who do not undergo antiviral treatment, such as recipients of solid organ and bone marrow transplants [52].
In this scenario, a valid inquiry emerges regarding whether employing antiretroviral drugs to downregulate HERV-K expression could potentially alleviate the malignant features exhibited by melanoma cells.
- Authors chose Lamivudine, Doravirine, and Cabotegravir, two reverse transcriptase and one integrase inhibitor. Authors should also explain integrase inhibitors in Intoduction that they only explained reverse transcriptase inhibitors. Even they should explain the difference of new non-nucleoside reverse transcriptase inhibitors or nucleoside reverse transcriptase inhibitors due to difference of lamivudine and doravirine.
- Authors should explain the potential of combinations of these drugs as they generally are used in combinations.
- What about their drawbacks in pharmacokinetic profiles? Is it safe to use these drugs?
- Why they choose these three drugs? According to chemical structure, safety,,,,? What are their superiorities?
Authors’ Reply:
We better clarify the different types and mechanisms of actions of the antiviral treatments, including the difference between nucleoside and non-nucleoside reverse transcriptase inhibitors, in the Introduction Section, as suggested. Then, we explained the potential of combinations therapy. Finally, we explained the rationale of the selected drugs (namely lamivudine as NRTI, doravirine as NNRTI and cabotegravir as INSTI), detailing their pharmacokinetics and safety (from line 127 to line 193 of the revised manuscript).
- Authors indicate that drugs to suppress the expression of HERV K Pol and Env genes. What about protease inhibition as it is included in Pol genes?
Authors’ Reply:
In the Introduction Section, we explained why protease inhibitors (PIs) have not been tested (lines 158-162):
“Despite their efficacy, Protease Inhibitors (PIs) have recently been discontinued as first-line antiretroviral therapy due to 'poor practicality', stemming from frequent interactions with concurrent medications and other issues. To choose a representative from each of the most frequently employed antiretroviral classes, we opted for lamivudine as an NRTI, doravirine as an NNRTI, and cabotegravir as an INSTI”.
In addition, data on the lower effects of PIs in in vitro studies was also reported in the Discussion Section: (lines 458-459)
“Tyagi et al. reported that protease inhibitors were less effective than RT or integrase inhibitors in hindering HERV replication [37].”
Round 2
Reviewer 1 Report
Comments and Suggestions for Authors
The authors have tried to take into account all the comments offered to them and have improved the manuscript and figures. As a small and subjective wish it may be noted that the font in the figures should be increased.
Author Response
Dear Reviewer,
as suggested, the font of the Figures has been enlarged.